# Acute Bronchitis and Bronchiolitis Infection in Children with Asthma and Allergic Rhinitis: A Retrospective Cohort Study Based on 5,027,486 Children in Taiwan

**DOI:** 10.3390/v15030810

**Published:** 2023-03-22

**Authors:** Fung-Chang Sung, Chang-Ching Wei, Chih-Hsin Muo, Shan P. Tsai, Chao W. Chen, Dennis P. H. Hsieh, Pei-Chun Chen, Chung-Yen Lu

**Affiliations:** 1Department of Health Services Administration, College of Public Health, China Medical University, Taichung 406, Taiwan; 2Management Office for Health Data, China Medical University Hospital, Taichung 404, Taiwan; 3Department of Food Nutrition and Health Biotechnology, Asia University, Taichung 413, Taiwan; 4Department of Pediatrics, College of Medicine, China Medical University, Taichung 404, Taiwan; 5Department of Public Health, College of Public Health, China Medical University, Taichung 406, Taiwan; 6School of Public Health, Texas A&M University, College Station, TX 77843, USA; 7University of Maryland Global Campus, Adelphi, MD 20783, USA; 8Department of Environmental Toxicology, University of California at Davis, Davis, CA 95616, USA; 9Department of Sport and Health Management, Da-Yeh University, Changhua 515, Taiwan; 10The School of Chinese Medicine for Post Baccalaureate, I-Shou University, Kaohsiung 824, Taiwan

**Keywords:** acute bronchitis and bronchiolitis, allergic rhinitis, asthma, human respiratory syncytial virus

## Abstract

This study evaluated the risks of childhood acute bronchitis and bronchiolitis (CABs) for children with asthma or allergic rhinitis (AR). Using insurance claims data of Taiwan, we identified, from children of ≤12 years old in 2000–2016, cohorts with and without asthma (N = 192,126, each) and cohorts with and without AR (N = 1,062,903, each) matched by sex and age. By the end of 2016, the asthma cohort had the highest bronchitis incidence, AR and non-asthma cohorts followed, and the lowest in the non-AR cohort (525.1, 322.4, 236.0 and 169.9 per 1000 person-years, respectively). The Cox method estimated adjusted hazard ratios (aHRs) of bronchitis were 1.82 (95% confidence interval (CI), 1.80–1.83) for the asthma cohort and 1.68 (95% CI, 1.68–1.69) for the AR cohort, relative to the respective comparisons. The bronchiolitis incidence rates for these cohorts were 42.7, 29.5, 28.5 and 20.1 per 1000 person-years, respectively. The aHRs of bronchiolitis were 1.50 (95% CI, 1.48–1.52) for the asthma cohort and 1.46 (95% CI, 1.45–1.47) for the AR cohort relative to their comparisons. The CABs incidence rates decreased substantially with increasing age, but were relatively similar for boys and girls. In conclusion, children with asthma are more likely to develop CABs than are children with AR.

## 1. Introduction

Childhood acute bronchitis and bronchiolitis (CABs) are respiratory inflammation conditions that most children may suffer from, and they are usually triggered by an infection characterized with seasonal meteorological conditions [1,2,3,4]. Bronchitis involves inflammation of bronchi mostly in older children and adults, whereas bronchiolitis involves inflammation of the bronchioles mainly for younger children and infants [4,5,6,7]. Both conditions are caused mainly by viruses, such as the flu virus, rhinovirus, respiratory syncytial virus (RSV) and bacterial pathogens, which can spread from person to person by droplets and through contact with an infected object [1,2,3,4,5,6,7,8]. Infections by RSV and influenza virus were recognized as the leading causes of CABs. The RSV infection can be implicated to children with bronchiolitis for up to 80% of cases [7], whereas the influenza virus infection is more common in patients with bronchitis. The seasonal infection generally peaks in the colder months because of more frequent indoor contacts and pathogen exposure when it is cold [3].

Respiratory disorders of other causes may also associate with the development of CABs. The relationships between asthma, bronchitis and bronchiolitis might be bidirectional [9,10,11,12]. More studies linked viral respiratory infections to the development of asthma [10,11,12,13,14]. Studies in Spain found children with HBoV-bronchiolitis, human metapneumovirus-bronchiolitis (hMPV), RSV-bronchiolitis or allergic rhinitis are at elevated risk of developing asthma at 5–7 years of age [10,11]. On the other hand, an earlier study in the US found that the severity of childhood bronchiolitis was associated with the severity of early childhood asthma [9].

Both asthma and allergic rhinitis are closely related to respiratory comorbidities and environmental conditions [13,14,15,16]. Similar to the seasonality of CABs, both asthma and allergic rhinitis may occur more often in colder months [17,18]. In fact, allergic rhinitis can also be associated with asthma, and asthma is often found in patients with allergic rhinitis [19,20]. However, studies evaluating whether children with asthma or allergic rhinitis are at higher risk of developing CABs is rare. This study aimed to assess risks of CABs between children with asthma and children with allergic rhinitis using insurance claims data of Taiwan.

## 2. Methods and Materials

### 2.1. Data Source and Study Population

This study used the national health insurance claims data of Taiwan for the period of 2000–2016 provided by the insurance authority. The data bank was available for data analysis at the Health and Welfare Data Science Center, which was designated by the authority. The insurance is a single-payer health care system providing universal mandatory coverage to all legal residents [21]. Over 99% of residents in Taiwan are covered. Information on socio-demographic status of insured individuals was available in the claims data, including sex, birthdate, income and occupation. Data on medical services and costs received were also available. Diseases were coded using the International Classification of Diseases, 9th Revision, Clinical Modification (ICD-9-CM) before 2016 and the International Classification of Diseases, 10 Revision, Clinical Modification (ICD-10-CM) since 2016. Individual identifications were re-encoded with surrogate numbers for data link to protect the patient privacy. The Research Ethics Committee of China Medical University and Hospital approved this study and waived individual patient consent (H107257). 

### 2.2. Establishment of Study Cohorts

From the claims data, we identified all children aged ≤12 years old in the period of 2000–2016 (Figure 1). Two sets of the study population were identified, one set consisted of children with and without asthma (ICD-9 cm code 493, ICD-10 cm code J45). Children who were diagnosed with asthma at least twice in outpatient records or once in the inpatient records were considered for the asthma group. After excluding children with the diagnosis of allergic rhinitis (ICD-9 cm code 477, ICD-10 cm code J30.9), we randomly selected an asthma cohort and a comparison cohort without the asthma diagnosis at a ratio of 1:1, frequency matched by sex and age. The other set of study population consisted of children with and without allergic rhinitis. Children who were diagnosed with allergic rhinitis at least twice in outpatient records or once in the inpatient records were considered for the allergic rhinitis group. After excluding children with the diagnosis of asthma, we randomly selected an allergic rhinitis cohort and a comparison cohort without the allergic rhinitis diagnosis at a ratio of 1:1, frequency matched by sex and age. All the four cohorts were followed until the diagnosis of acute bronchitis (ICD-9CM 466.0 and ICD-10 cm J20 and J20.9) or acute bronchiolitis (ICD-9 cm 466.11 and 466.19, and ICD-10 cm J21.0, J21.1, J21.8 and J21.9), or withdrawal from the insurance, or the end of 2016. The years of follow-up were estimated for each child.

### 2.3. Statistical Analysis

The baseline data on sex and age of children, and parental monthly income and occupation, were compared between cohorts with and without asthma, and between cohorts with and without allergic rhinitis. Incidence rates of acute bronchitis and acute bronchiolitis (CABs) were estimated for each cohort. The incidence rate was estimated as the number of incident cases divided by the sum of follow-up years for each cohort by variables of interest. The Cox proportional hazards regression models were used to estimate the asthma cohort to non-asthma cohort crude hazard ratio (cHR) and the adjusted hazard ratio (aHR) of CABs and related 95% confidence interval (CI). Similarly, the allergic rhinitis cohort to non- allergic rhinitis cohort cHR and aHR of CABs and related 95% CI were calculated as well. We used multivariable analysis to calculate the aHR after controlling for sex, age, parental income and other covariates.

## 3. Results

### 3.1. Baseline Characteristics

Our study population consisted of 192,126 children in each cohort with and without asthma and 1,062,903 children in each cohort with and without allergic rhinitis, with more boys in both sets of cohorts (53.2% and 54.5%, respectively) (Table 1). Asthma was more common in younger children, whereas allergic rhinitis was more common in older children. Both respective comparison cohorts were well matched by sex and age. However, the comparison cohorts had more children from lower income families than their respective asthma cohort and AR cohort.

### 3.2. Risk of Acute Bronchitis

Table 2 shows that the incidence of acute bronchitis was the highest in the asthma cohort, followed by the AR cohort (525.1 versus 322.4 per 1000 person-years) and the least in the non-AR comparisons (169.9 per 1000 person-years). The aHRs of acute bronchitis were 1.82 (95% CI = 1.80–1.83) for the asthma cohort and 1.68 (95% CI = 1.68–1.69) for the AR cohort, compared to the respective comparison cohort. The incidence of acute bronchitis was slightly higher in girls than in boys in each cohort, but the respective aHRs were similar for boys and girls in the asthma cohort and in the AR cohort. The incidence was higher in younger children and children from middle income families in each cohort, but the respective aHRs were similar among age groups and income groups.

### 3.3. Risk of Acute Bronchiolitis

Table 3 shows that the incidence of acute bronchiolitis was also the highest in the asthma cohort, followed by the AR cohort (42.7 versus 29.5 per 1000 person-years) and the least in the non-AR comparisons (20.1 per 1000 person-years). The aHRs of acute bronchiolitis were 1.50 (95% CI = 1.48–1.52) for the asthma cohort and 1.46 (95% CI = 1.45–1.47) for the AR cohort, compared to the respective comparison cohort. The incidence was higher for boys than for girls in the asthma cohort and their comparisons, whereas higher for girls than for boys in the AR cohort and their comparisons. The respective aHRs were somewhat alike for boys and girls. The incidence consistently decreased with increasing age in all four cohorts in somewhat similar patterns. Children in middle income families were found to have higher incidence.

## 4. Discussion

Most previous studies reported that patients with bronchitis or bronchiolitis of viral infection are at an increased risk of developing asthma [10,11,12,13,14]. A Swedish case–control study on infants with and without severe RSV bronchiolitis found that cases are at an elevated risk of developing allergic asthma persisting in 18-year-olds [12]. On the other hand, whether patients with asthma are at increased risk of CABs remains unclear. The risk of developing CABs in children with AR is even more obscure [18]. Our study might be the first population-based study assessing the risk of developing CABs for children with asthma or AR, with a large population size.

The possible mechanistic role for children with asthma or allergic rhinitis are more likely to develop CABs might be because their inflamed airways are weakened and sensitive to irritants, making it easier for viruses or bacteria to invade and cause infection. An earlier US study found 31% of infants with bronchiolitis had a history of asthma [9]. Additionally, children with asthma or allergic rhinitis may have weakened immune systems, making it more difficult for their bodies to fight off infections. This can also contribute to an increased risk of developing bronchitis or bronchiolitis.

The present study showed that children who were diagnosed with asthma or AR were at increased risk of developing CABs, with the incidence of bronchitis higher than that of bronchiolitis. The incidence rates of CABs were both greater in the asthma cohort than in the AR cohort, with the incident bronchitis and incident bronchiolitis 1.6-fold and 1.4-fold higher, respectively, in the asthma cohort than in the AR cohort. The data indicated that asthma is a stronger risk factor than AR in triggering the development of CABs and in exacerbating the bronchitis development. However, among 5,027,486 children identified from the claims data, the number of children diagnosed with AR was 1.7-fold more than children diagnosed with asthma (2,433,800 versus 1,405,100). The overall number of children developed CABs associated with AR was, thus, not less than that associated with asthma.

We reported, in a recent study, that the seasonality of CABs infection in children shows a parallel relationship with PM_2.5_ levels, but inversely relate to the monthly mean temperatures. The infection peaked in the coldest month, January, but dipped in February before climbing again in March [3]. We argued that the increased indoor activities and contacts among children in classrooms and daycare centers in cold months favored the pathogen spread. The dip of infection in February was because of lunar new year break with schools and daycare centers being closed. The CABs infection was the lowest in July, the hottest month in a year during the summer break. For the present study, we also analyzed the monthly incident asthma and AR in children by monthly average temperatures and PM_2.5_ levels (Appendix A). The seasonality model of the monthly incident asthma is similar to that of the of monthly CABs infection. The model of the monthly incident AR was also like that of CABs infection for most of the months except in July, in which the incidence climbed to 1.2-fold higher than the incidence in June (57 versus 46 per 100,000 persons). The seasonality models of asthma, AR and CABs reflect that these three disorders share some mutual allergens and likely similar childhood activities, but not in July for AR, when it is at the hottest month of the year in Taiwan. Apparently, it is not related to the air pollution in July because the PM_2.5_ level is the lowest in July.

In our study population, both asthma and AR cohorts consisted of more boys than girls. It is important to note that the HRs of CABs associated with asthma and AR are somewhat alike for boys and girls, suggesting no evidence of gender differences, consistently across all the four types of risk assessment. Yet, our data demonstrated sufficient linear age trends characterizing with large age variations. Both bronchitis and bronchiolitis incidence rates were much greater in children aged ≤2 years in all four cohorts. In the asthma cohort, the bronchitis incidence in children aged ≤2 years was 2.8-fold greater than that in 6–12 years old, with a difference of 496 per 1000 person-years, which was much greater than the difference (205 per 1000 person-years) between the two age groups of non-asthma comparison cohort. The incidence difference between these two age groups (504 per 1000 person-years) in the AR cohort was approximately similar to the finding in the asthma cohort.

The differences in incidence rates of bronchiolitis between the two age groups in both asthma cohort and AR cohort were similar, near 6-fold greater in the ≤2 years group than in the 6–12 years old, which were even greater than that of bronchitis. The younger children with asthma or AR were at an extremely higher risk than the older ones to develop bronchiolitis.

Our data also showed that the incidence of CABs tended to be higher in children from families with better incomes, particularly from the families with middle income. However, the HRs of CABs for the asthma cohort and the AR cohort, compared to the respective comparisons, varied little among income groups. We further found that the HRs of CABs for the asthma cohort and the AR cohort varied little among groups stratified by the urbanization level and parental occupation (Appendix A). These findings indicate that the impacts of asthma and AR on the risk of developing CABs are consistent.

The present study benefited by using a large population data, allowing comparing children with asthma and children with AR for the risk of developing CABs by stratified variables. The major study limitation was that we used the ICD codes to identify diseases based on physicians’ diagnoses. However, there were apparent contrasts in differences of CABs incidence rates between study cohorts and among age groups, reflecting that misclassification in identifying the study cohorts seemed minor and the bias might be neglectable. Second, the information on allergens and laboratory tests were unavailable in the claims data. We were able to use only ambient conditions and childhood activities to explain infection models. The allergen associated with AR appearing in the hot summer was not clear. Third, the risk of developing CABs in children of the asthma cohort may vary by asthma subtype. Similarly, the risk of developing CABs in children of the AR cohort may also vary by the subtype of AR. Further studies are warranted to examine these relationships. Fourth, it is unclear whether children who developed CABs are at an elevated risk of developing asthma or AR. Further studies are also needed to examine these relationships. Fifth, the observed relationship might be unique to the cohort of the children in Taiwan. For either genetic or environmental reasons, more studies of children outside this population are needed for generalization.

## 5. Conclusions

Previous studies showed that children with CABs are at elevated risk to trigger asthma and AR at later ages. Our data demonstrated these relations can be in the opposite direction. We found risks of developing CABs were greater in children with asthma than those with AR, particularly for the risk of developing bronchitis. However, the size of children with AR was near 2-fold more than the size of children with asthma. Children were at a much greater risk of developing bronchitis than getting bronchiolitis. Consistent with previous studies, children of ≤2 years old were at a much greater risk to have CABs than older children, which was similar between boys and girls. The seasonality patterns of asthma, AR and CABs suggest that the alertness and preventive measures for reducing the CABS risks in children with asthma and AR should be emphasized when it is cold.

## Figures and Tables

**Figure 1 viruses-15-00810-f001:**
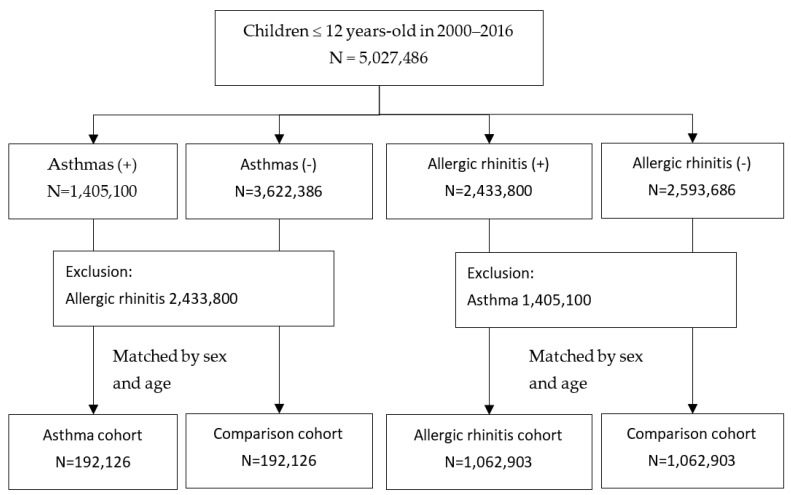
Flow chart for selecting cohorts with and without asthma and cohorts with and without allergic rhinitis from children of ≤12 years-old in 2000–2016.

**Table 1 viruses-15-00810-t001:** Comparison between cohorts with and without asthma and between cohorts with and without allergic rhinitis by demographic factors.

	**Asthma**				
	Yes (N = 192,126)		No (N = 192,126)		
Variable	N	%	n	%	*p*-value
Sex					1.00
Boy	102,182	53.2	102,182	53.2	
Girl	89,944	46.8	89,944	46.8	
Age, years					1.00
0–2	68,750	35.8	68,750	35.8	
3–5	79,811	41.5	79,811	41.5	
6–12	43,565	22.7	43,565	22.7	
Parental income					<0.0001
≤19,600 NTD	70,117	36.5	75,019	39.1	
19,601–27,600	56,651	29.5	56,337	29.3	
>27,600	65,358	34.0	60,770	31.6	
	**Allergic rhinitis**				
	Yes (N = 1,062,903)		No (N = 1,062,903)		
Variable	N	%	n	%	*p*-value
Sex					1.00
Boy	579,116	54.5	579,116	54.5	
Girl	483,787	45.5	483,787	45.5	
Age, years					1.00
0–2	258,528	24.3	258,528	24.3	
3–5	301,528	28.4	301,528	28.4	
6–12	502,847	47.3	502,847	47.3	
Parental income					<0.0001
≤19,600 NTD	407,253	38.3	474,196	44.6	
19,601–27,600	283,412	26.7	276,924	26.1	
>27,600	372,238	35.0	311,783	29.3	

**Table 2 viruses-15-00810-t002:** Incidence number and rate of acute bronchitis, asthma cohort to non-asthma cohort hazard ratio and allergic rhinitis cohort to non-allergic rhinitis cohort hazard ratio.

	**Asthma**					
	Yes (N = 192,126)		No (N = 192,126)		Hazard ratio (95% CI)	
Variable	n	Rate	n	Rate	Crude	Adjusted
All	166,527	525.1	143,977	236.0	1.78 (1.77–1.80)	1.82 (1.80–1.83)
Sex						
Boy	88,284	522.1	76,317	235.1	1.78 (1.76–1.79)	1.82 (1.80–1.83)
Girl	78,243	528.5	67,660	236.9	1.79 (1.77–1.81)	1.82 (1.80–1.84)
Age, years						
0–2	61,023	765.0	54,897	329.9	1.88 (1.85–1.90)	1.87 (1.84–1.89)
3–5	70,238	662.0	60,708	279.4	1.80 (1.78–1.82)	1.79 (1.77–1.81)
6–12	35,266	268.7	28,372	125.3	1.79 (0.77–1.82)	1.80 (1.77–1.83)
Parent income						
≤19,600 NTD	62,931	430.5	57,984	191.3	1.80 (1.78–1.82)	1.83 (1.81–1.85)
19,601–27,600	48,255	671.6	41,403	305.3	1.78 (1.76–1.80)	1.82 (1.80–1.84)
>27,600	55,341	558.6	44,590	260.0	1.76 (1.73–1.78)	1.80 (1.78–1.83)
	**Allergic rhinitis**					
	Yes (N = 1,062,903)		No (N = 1,062,903)		Hazard ratio (95% CI)	
	n	Rate	n	Rate	Crude	Adjusted
All	898,468	322.4	768,339	169.9	1.61 (1.60–1.61)	1.68 (1.68–1.69)
Sex						
Boy	487,218	310.6	414,159	163.7	1.61 (1.60–1.61)	1.68 (1.67–1.69)
Girl	411,250	337.6	354,180	177.8	1.61 (1.60–1.62)	1.68 (1.68–1.69)
Age, years						
0–2	232,433	706.6	208,599	324.0	1.77 (1.76–1.78)	1.78 (1.77–1.79)
3–5	267,786	540.4	232,491	247.3	1.67 (1.66–1.68)	1.67 (1.66–1.68)
6–12	398,249	202.9	327,249	111.4	1.62 (1.61–1.63)	1.63 (1.62–1.63)
Parent income						
≤19,600 NTD	354,006	280.5	349,285	145.0	1.65 (1.64–1.66)	1.74 (1.73–1.75)
19,601–27,600	234,364	400.6	197,344	218.2	1.57 (1.56–1.58)	1.66 (1.65–1.67)
>27,600	310,098	329.9	221,710	183.4	1.56 (1.55–1.57)	1.63 (1.62–1.64)

Rate, per 1000 person-years; CI, confidence interval.

**Table 3 viruses-15-00810-t003:** Incidence number and rate of acute bronchiolitis, asthma cohort to non-asthma cohort hazard ratio and allergic rhinitis cohort to non-allergic rhinitis cohort hazard ratio.

	**Asthma**					
	Yes (N = 192,126)		No (N = 192,126)		Hazard ratio (95% CI)	
Variable	n	Rate	n	Rate	Crude	Adjusted
All	54,222	42.7	41,365	28.5	1.45 (1.43–1.47)	1.50 (1.48–1.52)
Sex						
Boy	29,810	44.6	22,614	29.5	1.46 (1.44–1.49)	1.53 (1.50–1.55)
Girl	24,412	40.6	18,751	27.4	1.43 (1.41–1.46)	1.47 (1.45–1.50)
Age, years						
0–2	30,497	84.7	22,469	50.0	1.62 (1.59–1.65)	1.63 (1.60–1.66)
3–5	17,871	35.1	14,574	25.5	1.33 (1.30–1.36)	1.32 (1.29–1.35)
6–12	5854	14.6	4322	10.0	1.44 (1.38–1.50)	1.45 (1.39–1.51)
Parent Income						
≤19,600 NTD	19,169	30.7	14,672	19.7	1.51 (1.48–1.55)	1.56 (1.53–1.59)
19,601–27,600	16,754	61.9	13,381	43.4	1.39 (1.36–1.42)	1.47 (1.43–1.50)
>27,600	18,299	48.9	13,312	33.4	1.42 (1.39–1.45)	1.52 (1.49–1.56)
	**Allergic rhinitis**					
	Yes (N = 1,062,903)		No (N = 1,062,903)		Hazard ratio (95% CI)	
Variable	n	Rate	n	Rate	Crude	Adjusted
All	253,865	29.5	191,399	20.1	1.42 (1.41–1.43)	1.46 (1.45–1.47)
Sex						
Boy	135,734	28.5	101,670	19.3	1.42 (1.41–1.43)	1.47 (1.46–1.48)
Girl	118,131	30.8	89,729	21.1	1.41 (1.40–1.42)	1.45 (1.44–1.47)
Age, years						
0–2	113,139	84.5	91,262	56.8	1.41 (1.39–1.42)	1.40 (1.39–1.41)
3–5	71,732	32.5	52,692	21.3	1.46 (1.45–1.48)	1.47 (1.45–1.48)
6–12	68,994	13.6	47,445	8.74	1.54 (1.53–1.56)	1.56 (1.54–1.58)
Parent Income						
≤19,600 NTD	91,385	22.5	75,889	14.7	1.49 (1.47–1.50)	1.56 (1.55–1.58)
19,601–27,600	73,484	42.5	56,490	30.4	1.36 (1.35–1.38)	1.45 (1.43–1.46)
>27,600	88,996	31.7	59,020	23.6	1.33 (1.31–1.34)	1.38 (1.36–1.39)

Rate, per 1000 person years; CI, confidence interval.

## Data Availability

Data were retrieved from the insurance claims data of Taiwan (http://nhird.nhri.org.tw/, accessed on 9 October 2019) at the Health and Welfare Data Science Center: and access to this database can be requested by sending a formal proposal to the NHI.

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
