# Peer review of "Acute Bronchitis and Bronchiolitis Infection in Children with Asthma and Allergic Rhinitis: A Retrospective Cohort Study Based on 5,027,486 Children in Taiwan"

_viruses, 2023, doi:10.3390/v15030810_

Round 1

Reviewer 1 Report

Sung et al. presented an interesting retrospective cohort study where they analysed the risk of childhood acute bronchitis or bronchiolitis in children with asthma or allergic rhinitis. The strength of the study was the availability of data from a large cohort, and intriguingly the hypothesis was that relation between childhood aunt bronchitis/bronchiolitis and asthma/allergic rhinitis may not be as unidirectional as previously described.

Nevertheless, the reviewer has the following concerns and curiosities regarding the manuscript:

1) In the methods section it is not really clear which kind of asthma has been taken in consideration. Assuming the stratification did not take in consideration the different asthma types (mild/moderate/severe, allergic/non-allergic ecc...), would it be possible to retrieve this information in order to understand whether the incidence of bronchitis/bronchiolitis is higher in one asthma subtype rather than in others, or if this is general?

2) Similarly, history of allergic rhinitis is not stratified. Since allergic rhinitis can be triggered by different factors (mold, pollens, animal dander, dust) it may be inferred that one or the other environmental factor can impact the incidence of bronchitis/bronchiolitis more than others. Further, environmental factors may further explain the higher incidents/rate in one or the other parental income class. Can the authors provide more info in this regard?

3) According to the method section, the cohorts were followed until the diagnosis of acute bronchitis or acute bronchiolitis. However, a complete picture would have required for all those children to be followed even after the diagnosis of bronchitis/bronchiolitis. This would have been especially important for the comparison cohorts, which may have shown cases of asthma/rhinitis development following the airway disease occurrence. Can the authors retrieve this information in order to have the complete picture?

4) Based on point 3, it appears to the reviewer that the authors conclusions regarding the cause/effect between CABs and asthma/AR is opposite direction with respect to what has been shown in other studies need to be somehow quenched. It is indeed true that, as shown in this study, asthma and AR may cause CABs, however this should not esclude the opposite direction. It is reviewer's opinion that both cause/effect should be interconnected within the two sets of airway diseases, and both directions are equally valid. Can the authors elaborate a bit more about this?

Reviewer 2 Report

This is a relatively simple paper by Sung et al. that explores the correlation, if any, between childhood acute bronchitis and bronchiolitis (CABs) and asthma or allergic rhinitis (AR). The authors' team used datasets of Taiwanese pediatric population, obtained from the health insurance agencies. This work is a typical meta-analysis of patient health data, and are reasonably conducted and clearly presented. The Introduction should be particularly useful to the general readers. Interestingly, and perhaps, counterintuitively, the analysis revealed that the CAB is greater in children with asthma than those with AR.

 The authors should state the limitations of the study, such as the observed relationship may be unique to the cohort of the children in Taiwan, for either genetic or environmental reasons and that future studies of children outside this nation will be needed for generalization.

 It would also be useful if a mechanism is suggested for the relationship, even if speculative. This can go in the Discussion section.

Minor: English needs improvement / correction in several places.
